

# New chaos-integrated improved grey wolf optimization based models for automatic detection of depression in online social media and networks

Sinem Akyol

Software Engineering Department, Engineering Faculty, Firat University, Elazığ, Turkey

Corresponding author
Sinem Akyol, sakyol@firat.edu.tr

## ABSTRACT

Depression is a psychological effect of the modern lifestyle on people's thoughts. It is a serious individual and social health problem due to the risk of suicide and loss of workforce, high chronicity, recurrence rates, and prevalence. Therefore, identification, prevention, treatment of depression, and determination of relapse risk factors are of great importance. Depression has traditionally been diagnosed using standardized scales that require clinical diagnoses or patients' subjective responses. However, these classical techniques have some limitations such as cost, uncomfortability, subjectivity, and ineffectiveness. Social media data can be simply and efficiently used for depression detection because it allows instantaneous emotional expression and quick access to various information. Some machine learning-based methods are used for detecting the depression in online social media and networks. Nevertheless, these algorithms suffer from several drawbacks, including data sparsity, dimension explosion, restricted capacity for generalization, and low performance on imbalanced data sets. Furthermore, many machine learning methods work as black-box models, and the constructed depression detection models are not interpretable and explainable. Intelligent meta-heuristic optimization algorithms are widely used for different types of complex real-world problems due to their simplicity and high performance. It is aimed to remove the limitations of studies on this problem by increasing the success rate and automatically selecting the relevant features and integrating the explainability. In this study, new chaos-integrated multi-objective optimization algorithms are proposed to increase efficiency. New improved Grey Wolf Optimization algorithms have been proposed by integrating Circle, Logistic, and Iterative chaotic maps into the improved Grey Wolf Optimization algorithm. It is aimed to increase the success rate by proposing a multi-objective fitness function that can optimize the accuracy and the number of features simultaneously. The proposed algorithms are compared with different types of popular supervised machine learning algorithms and current metaheuristic algorithms that are widely and successfully used in depression detection problems. Experimental results show that the proposed models outperform machine learning methods, as evidenced by examining results with accuracy, F-measure, MCC, sensitivity, and precision measures. An accuracy value of 100% was obtained from proposed algorithms. In addition, when the confusion matrices are examined, it is seen that they exhibit a successful distribution. Although it is a new research and application area for optimization theory, promising results have been obtained from the proposed models.

# INTRODUCTION

Depression is defined as a mental disorder that appears in the form of decreased sensitivity to stimuli, loss of initiative and self-confidence, and strengthening of hopelessness and pessimism. Depression and emotional distress are psychological effects of the modern lifestyle on people's thoughts. Depression can begin at any age, however, the World Health Organization reports that most suicides caused by depression occur in people between the ages of 15 and 29. Sleep disturbances, a lack of energy, a lack of feelings of worthlessness, interest in daily activities, inability to concentrate, and recurrent suicidality are all symptoms of depression (*Liu et al., 2022*). High levels of depression, a mental disease characterized by a low mood and a reluctance to participate in activities, have a significant impact on people's thoughts and behavior (*Zhao et al., 2023*). Depression detection is critical for assisting in the relief of these threats. There are numerous reasons for depression, some of which humans do not fully comprehend. Figure 1 depicts seven of the widely accepted types of depression (*Schimelpfening, 2023*). Their characterizations with key features are also listed in this figure.

Identification, prevention, treatment of depression, and determination of relapse risk factors are of great importance as it is a serious individual and social health problem due to the risk of suicide and loss of workforce, high chronicity, and recurrence rates and prevalence (*Amanat et al., 2022*). Depression can have a negative impact on individual's socioeconomic status. People who are depressed are less likely to socialize. Counseling and psychological therapies can aid in treating depression. Due to people's disabilities in various situations, depression is a big concern on a global scale. Depression, which is a single mental health illness with a variety of diagnostic techniques, is known to cause major social issues in people, including suicide (*Gupta & Sharma, 2021*). Consequently, addressing the burden of mental health concerns calls for a comprehensive approach.

Depression has traditionally been diagnosed using standardized scales that require clinical diagnoses or patients' subjective responses provided by attending clinicians, and these classical techniques have some limitations. First, people's responses to traditional standardized scales are probably influenced by the patient's mental state at the time, context, patient's current mood, the clinician-patient relationship, and the patient's memory bias and past experiences. Classical diagnostic methods are also limited in terms of temporal granularity. Interviews and subjective diagnostic criteria often lead to ambiguous and contradictory diagnoses. The second limitation is that individuals may be ashamed or unaware of their depressive symptoms and are less likely to seek professional help, particularly in the early stages. According to the study (*Shen et al., 2017*), if the patient was in the early stages of depression, more than 70% of the population would not seek professional help, implying that they would likely wait for their symptoms to worsen before needing support. Finally, traditional methods of detecting depression, which rely

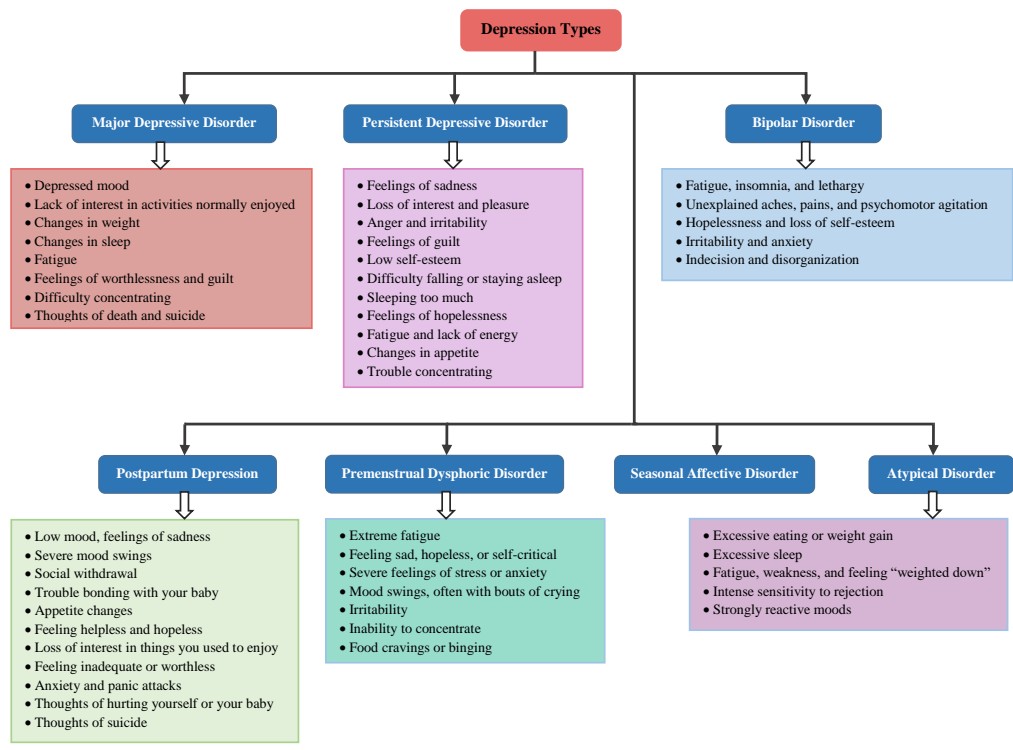

**Figure 1** Depression types.

on face-to-face interviews, are expensive concerning both time and money, making them pricey for some people (*Liu et al., 2022*). The conventional clinical diagnosis for depression also does not provide an effective or systematic method for integrating behavioral findings that indicate the severity and presence of depression. As a result, more cost-effective methods for detecting depression in large populations are required.

The resources available to recognize and manage states of depression or suicidality are typically insufficient. Online data and records are increasingly being recognized as viable sources of information for public health decisions. By the way, social media has developed into a medium for people to express their feelings, opinions, ideas, and emotions, which makes them feel satisfied to share ideas that are upsetting to them. People's use of social media has evolved and created trends, and due to the social network's growing popularity, more people are using it to publicly express themselves online. Adolescent use of digital and social media. Previous research has found that has changed the way they form and maintain friendships. While social media allows early teens to form new relationships online, providing them with the opportunity to connect with their peers, it also creates new obligations, expectations, and social dynamics that result from continued commitment to friendships. Persistent connection and the social norms that govern digital media can cause stress and strain on friendships (*De Groote & Van Ouytsel, 2022*). In addition, studies show that the higher the fear of depression, the stronger the habit of using and constantly

checking social media. It shows that this can eventually trigger emotional and psychological states such as depression (*Park, 2022*).

Although questionnaires and interviews have historically been used in psychological research, many researchers are increasingly collecting data online. They seek to use the methods they provide to psychologically analyze social network data and draw conclusions. Using web text shared by individuals on social media, one can extract the emotions and expressions hidden within the text. Researchers have used data obtained from social media (posts, tweets, size of social network, likes, comments, join groups demographics, status updates, etc.) as indicators for online diagnosing the situation of users, such as whether or not the user suffers from depression (*Latif et al., 2021*). Because social media allows instantaneous emotional expression and easy access to a wealth of information, recognizing depression is now an important step in reducing the death rate (*Aleem et al., 2022*).

With rapid advances in machine learning approaches, it is critical to incorporate modern technology into depression diagnosis to prevent depressive individuals from committing suicide through earlier and more accurate depression detection. The classical clinical diagnostic method for depression does not appropriately identify the complexity of depression. Using machine learning methods, the symptom composition associated with mental disorders such as depression can be easily predicted and detected. As a result, the machine learning-based diagnostic approach appears to be a viable option for predictive analysis. Text, sensors, structured data, and multimodal technology interactions are the major domains used in the healthcare sector for extracting observations associated with mental disorders using machine learning (*Aleem et al., 2022*). Text messages, social media platforms, and clinical records can all be used to extract text sources. Audio signals and mobile phones can be used to analyze sensor data. Data extracted from traditional screening questionnaires, medical health records, and scales comprise the structured data. Data from human interactions with everyday technological equipment, virtual agents, and robots are included in the multimodal technology interactions.

Previous research has found that automatic text-based analysis of depressive symptoms can be used to detect depressing phrases or disrespectful sentences in blog posts or conversations as well as sentiment recovery from suicide notes (*Liu et al., 2022*). The classification of shared posts in blogs is critical to investigate to save desperate people's lives. Fortunately, applying machine learning specifically to text data from social media can provide a viable solution to the problem of depression detection. Social media platforms such as Facebook, Twitter, microblogs, and discussion forums have long been popular for expressing and documenting people's thoughts, personalities, feelings, behaviors, and moods (*Liu et al., 2022*). However, research into extracting depression symptoms from text remains promising. Machine learning has revolutionized latent knowledge extraction in the field of text processing tending to understand the context of complex natural language sentences (*Gupta & Sharma, 2021*). Existing machine learning-based depression detection systems go through a series of steps, including preprocessing, feature extraction, and depression detection. Existing depression diagnostic research (*Goodarzi et al., 2017*) has discovered several flaws in accurately detecting depression in a timely manner. Traditional

research has acquired accurate information about depression-indicative symptoms to effectively identify and diagnose depression from user-generated natural language text. Even though it is a crucial task, extracting the depression-indicative symptoms from violation-rich social media material results in erroneous depression detection. Therefore, to enhance the effectiveness of depression identification in social media, numerous artificial intelligence depression detection models have employed feature extraction and data preprocessing techniques to the raw input textual messages to produce a potential feature set for the learning model (*William & Suhartono, 2021*; *Kim, Lee & Park, 2021*; *Wongkoblap, Vadillo & Curcin, 2021*; *Figueredo & Calumby, 2020*; *Lin et al., 2020*; *Tao et al., 2016*). Extensive monitoring of online social networks and the use of machine learning methods on data obtained from social media can automatically diagnose depressive symptoms, helping identify individuals at risk of depression and supporting current screening methods.

Although some machine learning-based methods applied to social media data have the potential as a way to identify people at depression risk through large-scale monitoring of social media and could complement classical screening methods, some methods achieved poor results in terms of different metrics. Furthermore, many machine learning-based methods suffer from some drawbacks, including data sparsity, dimension explosion, restricted capacity for generalization, and low performance on imbalanced data sets. Additionaly, many machine learning methods work as black-box models, and the constructed depression detection models are not interpretable, explainable, and comprehensible. To increase the success rate and automatically integrate explainability for depression detection in online social media and networks, new chaos-integrated optimization algorithms are proposed in this article. Chaos-integrated improved Grey Wolf Optimizer (GWO) can also be used as a general solution search and optimization algorithm for different types of complex problems.

A study in which chaotic maps were adapted to metaheuristic algorithms was conducted by *Bacanin et al. (2021)*. This study examines the performance of an improved chaotic firefly algorithm and tests its ability to automatically select the optimal dropout rate in deep learning applications. Theoretical and practical experiments show that the proposed algorithm performs better than other methods. This work represents an important development in the field of deep learning that can contribute to the prevention of overfitting (*Bacanin et al., 2021*). *Malakar et al. (2020)* introduce a Genetic Algorithm-based hierarchical feature selection model for handwritten word recognition. They used this model to optimize local and global features from handwritten word images. They tested it on a dataset containing 12,000 samples of handwritten words written in Bangla script. With the proposed model, better results were obtained compared to existing methods by reducing the feature size by 28% while increasing the word recognition performance by 1.28% (*Malakar et al., 2020*). In these two studies, it is seen that the new research field successfully combines machine learning and swarm intelligence approaches and can achieve extraordinary results in different fields.

To emphasize the effects of depression on people's mental health in modern life and the seriousness of this problem. The high risk of suicide and job loss, chronicity, relapse rates, and high prevalence show that depression is a major problem in terms of individual

and social health. Therefore, it is important to diagnose, prevent, and treat depression and identify relapse risk factors. The limitations and deficiencies of traditional diagnostic methods require the investigation of new approaches to solve this problem. Social media data is seen as a useful resource as part of these new approaches, but existing methods in this area have limitations and challenges. In this context, overcoming the limitations in detecting and managing depression by using intelligent meta-heuristic optimization algorithms is the motivation of this study.

The main contributions of this study are listed below:

- Social media and network data are considered as problem search space, and the new improved optimization methods proposed in this study are adapted as solution search strategies and models for automatic depression detection.
- New explainable, comprehensible, and interpretable depression detection models different from the classical black box algorithms are proposed.
- The new improved chaos-integrated general purpose GWO algorithms are proposed as a global optimization and solution search methodology.
- To improve efficiency, chaos theory is embedded in the optimization method and a flexible fitness function is created for the intelligent optimization algorithm. The fitness function is easily adaptable to different goals and can be efficiently added to the fitness computation.
- The proposed new solution search methodologies, which include appropriate encoding type, flexible multi-objective fitness function, and chaos theory, can be easily applied to a variety of social network and media problems.
- A new area of research and application is addressed for optimization theory.
- Despite being an innovative approach, the proposed models yield encouraging outcomes.

The article is organized as follows: Related works are reviewed in the Related Works section. The theoretical background on GWO and chaotic maps is presented in the Theoretical Works section. The proposed general purposed chaos-integrated new improved GWO (NI-GWO) algorithms and adaptation of these algorithms for the problem of depression detection are described in the Proposed Works section. The Experimental Results section presents the experimental results. Finally, the article is concluded and possible future works are mentioned in the Conclusions.

## RELATED WORKS

There is currently no effective clinical description of depression. It limits and biases the diagnostic procedure. The difficulty of diagnosing depression depends not only on the patient's educational level, cognitive aptitude, and honesty in describing their symptoms but also on the physicians' experience and motivation. To effectively diagnose the degree of depression, much clinical expertise and knowledge are required (*Aleem et al., 2022*). In order to automatically evaluate the severity scale of depression, various automatic depression estimating methods have been established in recent years. A brief categorization of related works on depression detection is depicted in Fig. 2. Recent discussion forums
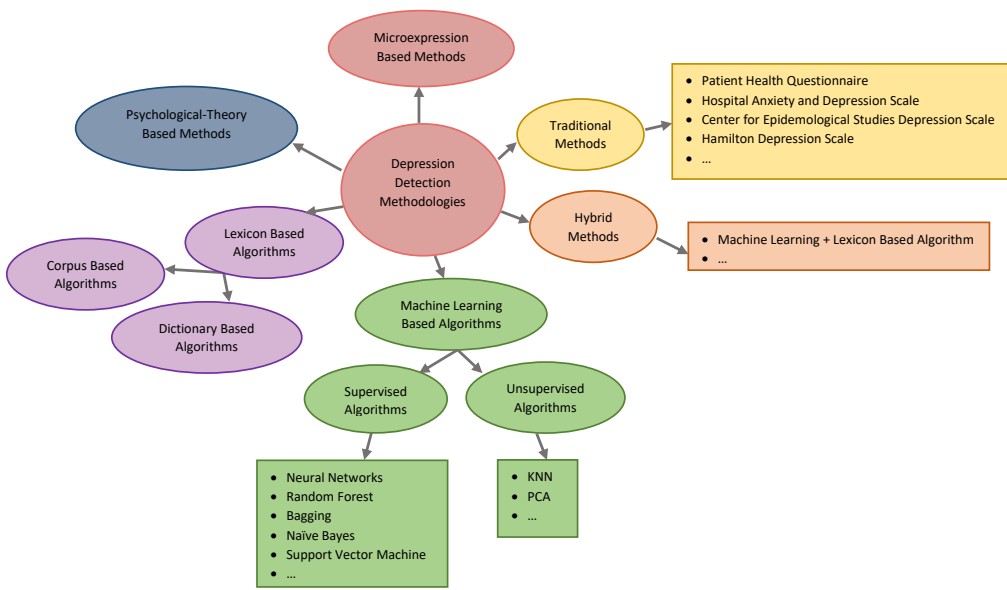

**Figure 2    A brief categorization of related works on depression detection.**

have concentrated in particular on identifying depressed individuals through the analysis of their social media posts. In this section, related works about depression detection methods, models, and systems in online social media and networks are presented.

The effects of employing a supervised machine learning method to assess predictors for detecting post-traumatic severe depression were covered by *Kim, Lee & Park (2021)*. As part of their research, they used Twitter users to create a model for each linguistic style. A deep neural network approach was put forward by *Wongkoblap, Vadillo & Curcin (2021)* to examine depression in social media. For their analysis, they used the Twitter dataset.

With the aid of preprocessing techniques, the depression detection model (*Figueredo & Calumby, 2020*) enhances the accuracy of the prediction. The textual data preparation for depression identification aids in mapping emotions to actual emotion words, which maximizes the early and highly relevant detection of depression. Through a deep visual-textual multimodal learning method, the Sense Mood model (*Lin et al., 2020*) efficiently analyses and detects users' depression. Based on an analysis of the urgent demand for methods, the sentiment analysis-based depression detection approach (*Tao et al., 2016*) detects potential depression early. It determines the depressive people using the analysis of the psychological status of the users in the social network by analyzing the user-generated content in the social network through sentiment analysis without compromising the privacy of the users in the social network. To prevent suicide, the study (*Leiva & Freire, 2017*) sequentially analyzes user posts in a social network to detect the risk of depression early on. Furthermore, it penalizes the delay in detecting depressive people by using a combination of learning algorithms to recognize early depression symptoms.

By analyzing clinical depression and online behaviors from various social media perspectives, the multimodal dictionary learning approach (*Shen et al., 2017*) extracts

six depression-related attribute groups. It detects depressed users in the Twitter network in real-time using well-labeled depression and nondepression data, assisting in proactive care for depressed social users. The study *Trotzek, Koitka & Friedrich (2018)* uses a convolutional neural network and various word embeddings to detect depression from user-level linguistic metadata analysis. Using the learning model, the indications of depression in written texts generated by social users are classified.

The researchers proposed a lexicon-based approach for detecting depression (*Li et al., 2020*). Word2Vec, the label propagation algorithm, and a semantic relationship graph were used to create a lexicon. The authors based their findings on 111,052 Weibo microblogs consisting of 1,868 depressed or non-depressed users. To predict depression, they used six features and five classification methods. The results show that the generated lexicon performed better for classification.

Most previous studies had different limitations in terms of test dataset size, text analysis languages, single platform, and levels of depression, which aid psychiatrists in following treatment steps according to the detected level. Social media text analysis can develop methods for early detection of associated ailments and for determining the mental health status of users (*Trifan et al., 2020*).

Many methods for depression detection in online social network messages either use person descriptive-based highlighting approaches or textual-based highlighting approaches. The linguistic features of social network text, such as words, n-grams, part-of-speech, and other linguistic characteristics, are the focus of textual-based features (*Chiong, Budhi & Dhakal, 2021a*; *Chiong et al., 2021b*). In contrast, the descriptive-based featuring methodology focuses on subject descriptions such as employment status, age, gender, smoking, alcohol or drug consumption, income, and other details about the patient or subject (*De Souza Filho et al., 2021*). However, it can be concluded that people's actions on social media have multiple dimensions. Understanding the interaction through various modalities is difficult, as is characterizing people from selective viewpoints. Table 1 summarizes the literature works about depression detection in online social media and networks along with the limitations.

## THEORETICAL BACKGROUND

Intelligent metaheuristic optimization algorithms are widely used for different types of complex real-world problems due to their simplicity and high performance (*Akyol, 2022*). Many problems can be modeled as an optimization problem, and metaheuristic methods can be easily adapted as solution search strategies for the focused problem. These algorithms are categorized into nine different classes based on the inspiration sources (*Akyol & Alatas, 2017*). Specifically, biology-based and swarm-based metaheuristic methods are preferred for the search and optimization problem. In this article, a swarm-based GWO algorithm is adapted as a base for an automatic and direct solution search strategy for the focused depression detection problem.

**Table 1  Summary of literature works on depression detection in online social networks.**

| Study | Data Source | Methods | Results | Limitations |
|---|---|---|---|---|
| *AlSagri & Ykhlef (2020)* | Twitter | Naïve Bayes, Support Vector Machines, Decision Tree | Accuracy: 0.825<br>Precision: 0.739<br>Sensitivity: 0.850<br>F-measure: 0.791<br>AUC: 0.780 | Utilizing the different kernels in SVM<br>Not avoiding overfitting dataset<br>Lack of interpretability and comprehensibility |
| *Kim et al. (2020)* | Reddit | XGBoost, Convolutional Neural Network | Accuracy: 0.751<br>Precision: 0.891<br>Sensitivity: 0.718<br>F-measure: 0.795 | Low success rate<br>Lack of explainability<br>Limited scope |
| *Fatima et al. (2018)* | LiveJournal | Random Forest | Accuracy: 0.918 | Small size of data<br>Single machine-learning algorithm<br>Absence of other evaluation metrics<br>Inefficiency in unbalanced dataset |
| *Orabi et al. (2018)* | Twitter | Convolutional Neural Network, Bidirectional LSTM | Accuracy: 0.850 | Small size of data<br>Lack of interpretability and comprehensibility<br>Hardness task of exercising for RNN<br>Report and Slope disappearing problems |
| *De Choudhury et al. (2013)* | Twitter | Support Vector Machines | Accuracy: 0.700 | Low success rate<br>Lack of explainability<br>Single machine-learning algorithm<br>Inappropriateness for large data sets |
| *Thorstad & Wolff (2019)* | Reddit | Cluster Analysis, Logistic Regression | Accuracy: = 0.390<br>F-measure = 0.380 | Low performance<br>Lack of interpretability |
| *Nadeem (2016)* | Twitter | Support Vector Machines, Decision Tree, Naïve Bayes, Logistic Regression | Accuracy: 0.860<br>Sensitivity: 0.830<br>F-measure: 0.840 | Usage of the old dataset<br>Emphasis on user confession<br>Lack of interpretability |
| *Aldarwish & Ahmad (2017)* | Facebook, Twitter, and LiveJournal | Support Vector Machines, Naïve Bayes, | Accuracy: 0.633<br>Sensitivity: 0.570 | Usage of old Arabic dataset<br>Limited phrases and sentences<br>Low success rate |
| *Gaikar et al. (2019)* | Twitter | Support Vector Machines–Naïve Bayes hybrid model | Accuracy: 0.850 | High computational complexity for comparing the long-short snippets.<br>Need for determining the appropriate values for combined methods<br>Lack of interpretability |
| *Islam et al. (2018)* | Facebook | Support Vector Machines and LIWC | Accuracy: 0.700 | Inappropriateness for large data sets<br>Lack of interpretability |
| *Wang et al. (2018)* | Reddit | Convolutional Neural Network | F-measure: 0.670 | Not encrypting the situation and alignment of an entity<br>Lack of interpretability |
**Table 1** (*continued*)

| Study | Data Source | Methods | Results | Limitations |
|---|---|---|---|---|
| *Burdisso, Errecalde & MontesyGómez (2019)* | Reddit | SS3 | F-measure: 0.610<br>Precision: 0.630<br>Sensitivity: 0.600 | Small size of data<br>Low performance<br>Lack of interpretability |
| *Adarsh et al. (2023)* | Reddit | One-shot Decision, Combining of SVM and KNN | Accuracy: 0.981<br>Precision: 0.968<br>Sensitivity: 0.976<br>F-measure: 0.973<br>AUC: 0.979 | Lack of handling multiclass depression classification<br>Requiring too many parameters that should be adjusted a priori |
| *Gupta, Pokhriyal & Gola (2022)* | Reddit | Combining Convolutional Neural Network and LSTM | Accuracy: 0.940<br>Precision: 0.942<br>Sensitivity: 0.937<br>F-measure: 0.940 | Necessity of many values that should be adjusted a priori for the combined methods<br>Lack of explainability<br>Inappropriateness for unbalanced data sets |
| *Chen et al. (2023)* | Reddit | Combining Convolutional Neural Network and SBERT | Accuracy: 0.860<br>Precision: 0.850<br>Sensitivity: 0.870<br>F-measure: 0.860 | High computational cost<br>Lack of explainability<br>Requiring too many parameters |

## Grey wolf optimizer

The grey wolf optimizer (GWO) is a swarm-based metaheuristic method inspired by the hunting skills and leadership hierarchies of gray wolves within the swarm. There are four types of wolves in the GWO method, which is inspired by the hunting behavior of gray wolves and their social leadership in nature. The $\alpha, \beta$, and $\delta$ wolves have the best position, respectively, and the $\omega$ wolves, who are out of these wolves and will be led by these wolves. Hunting of wolves consists of three stages: encircling, hunting, and attacking the prey (*Mirjalili, Mirjalili & Lewis, 2014*).

Encircling: It is the stage of encircling the prey by gray boxes and is expressed by Eqs. (1) and (2).

$$D = |C \times X_p(t) - X(t)| \tag{1}$$

$$X(t+1) = X_p(t) - A \times D \tag{2}$$

$X_p$ represents the position of the prey, $D$ represents the distance between the current candidate wolves and the top three wolves, $t$ represents the current iteration, $X$ is used for the position vector of a gray wolf, and the coefficients $A$ and $C$ are the coefficients computed by the equations in Eqs. (3) and (4) (*Mirjalili, Mirjalili & Lewis, 2014*).

$$A = 2 \times a(t) \times r_1 - a(t) \tag{3}$$

$$C = 2 \times r_2 \tag{4}$$

The element values of vector $a$ are decremented from 2 to 0 each time the algorithm is run. $r_1$ and $r_2$ are random vectors that take values in the range $[0, 1]$.

Hunting: At this stage, which mathematically models the hunting behavior of wolves, the $\omega$ wolves follow the $\alpha, \beta$, and $\delta$ wolves, which are assumed to know the location of the prey best. The equations for the hunting phase are given in Eqs. (5)–(7) (*Nadimi-Shahraki, Taghian & Mirjalili, 2021*).

$$
\begin{aligned}
D_\alpha &= |C_1 - X_\alpha - X(t)| \\
D_\beta &= |C_2 - X_\beta - X(t)| \\
D_\delta &= |C_3 - X_\delta - X(t)|
\end{aligned}
\tag{5}
$$

$C_1$, $C_2$, and $C_3$ are calculated using Eq. (4).

$$
\begin{aligned}
X_{i1} &= X_\alpha(t) - A_{i1} \times D_\alpha(t) \\
X_{i2} &= X_\beta(t) - A_{i2} \times D_\beta(t) \\
X_{i3} &= X_\delta(t) - A_{i3} \times D_\delta(t)
\end{aligned}
\tag{6}
$$

$X_\alpha$, $X_\beta$, and $X_\delta$ are the three best candidate solutions in the $t$ iteration. $A_1$, $A_2$, and $A_3$ are computed using Eq. (3).

$$
X(t+1) = \frac{X_{i1}(t) + X_{i2}(t) + X_{i3}(t)}{3}
\tag{7}
$$

Attacking: The process of hunting wolves ends when the prey terminates moving. After this point, the attacking process begins. This process is mathematically expressed as a decrease in the value of $a$ from 2 to 0 over the iterations. According to *Emary, Zawbaa & Grosan (2017)*, half of the iterations are for exploration and the other half for exploitation. In the attacking step, the wolves move their position to an arbitrary position between the prey position and their current position.

In the algorithm, first, the wolf population placed at random locations in the search area is created and the fitness values are calculated according to the locations of these wolves. The algorithm is terminated when it reaches the *MaxIter* number. In each iteration, the encircling, hunting, and attack steps are repeated. $a$, which represents the best position of the prey, is the solution. The improved GWO (I-GWO) algorithm has been proposed because of the risks of early convergence, the imbalance between exploration and exploitation, and the population diversity of the standard GWO algorithm (*Nadimi-Shahraki, Taghian & Mirjalili, 2021*).

### Improved grey wolf optimizer

In the standard GWO; the $\alpha, \beta$, and $\delta$ wolves with the best positions direct the remaining $\omega$ wolves to promised areas in the search space, and the decrease in diversity of population may cause the GWO to be stuck at the local optimum. To overcome these problems, a new search strategy is associated with the update and select step in the proposed I-GWO. I-GWO includes initialization, movement, and selection and update phases.

Initializing stage: In this stage, random $N$ wolves are created in the search space, as shown in Eq. (8) (*Nadimi-Shahraki, Taghian & Mirjalili, 2021*).

$$
X_{ij} = l_j + rand_j[0,1] \times (u_j - l_j), i \in [1, N], j \in [1, M]
\tag{8}
$$

$M$ represents the size of the problem, $l_j$ represents the lower limit that the $j$ th decision variable can take, and $u_j$ represents the upper limit. $rand_j$ generates a random number

between 0 and 1. The fitness value of $X_i(t)$, which represents the position of the $i$th wolf in the $t$th iteration, is evaluated according to the fitness function.

Movement stage: In addition to the group hunting strategy, a dimensional learning-based hunting (DLH) strategy was added for individual hunting at this stage. In the DLH stage, each wolf is learned by neighboring wolves as another candidate for the newly available position of $X_i(t)$ (*Nadimi-Shahraki, Taghian & Mirjalili, 2021*).

Canonical GWO search strategy: After determining the $\alpha, \beta,$ and $\delta$ wolves with the best fitness values in the population, the linear decreasing coefficient $a$ and the coefficients $A$ and $C$ are computed by Eq. (3)–(5). As shown in Eqs. (5) and (6); $X_\alpha, X_\beta,$ and $X_\delta$ are used to find the surrounding prey. Finally, Eq. (8) is used to calculate the new position of $X_i(t)$.

DLH search strategy: In the GWO, location updates are made based on the three lead wolves in the population. This causes the algorithm to converge slowly and reduces the diversity in the population. To overcome these weaknesses of the algorithm, the proposed DLH search strategy considers other wolves in the population when updating the location. The position of the wolf $X_i(t)$ in the DLH search strategy is calculated using Eq. (11). When updating the location, a randomly selected wolf from the population and different neighbors are used. In this strategy, in addition to $X_{i-GWO}(t+1)$ for the wolf $X_i(t)$, another candidate solution, $X_{i-DLH}(t+1)$ is produced. First, the value of $R_i(t)$, which is the Euclidean distance between the candidate solution $X_i(t)$ and $X_{i-GWO}(t+1)$, is calculated using Eq. (9) (*Nadimi-Shahraki, Taghian & Mirjalili, 2021*).

$$R_i(t) = \|X_i(t) - X_{i-GWO}(t+1)\| \tag{9}$$

Then, the neighbors of $X_i(t)$ denoted by $N_i(t)$ are found using Eq. (10) depending on the radius of $R_i(t)$.

$$N_i(t) = \{X_j(t) | D_i(X_i(t), X_j(t)) \le R_i(t), X_j(t) \in Pop\} \tag{10}$$

Here, $D_i$ represents the distance between $X_i$ and $X_j$. After the neighborhood of $X_i(t)$ is created, and multineighbor learning is calculated using Eq. (11). The $d$th dimension of $X_{i-DLH}(t+1)$ is calculated using a randomly selected neighbor $X_{n,d}(t)$ from $N_i(t)$ and randomly selected $X_{r,d}(t)$ from the population.

$$X_{i-DLH,d}(t+1) = X_{i,d}(t) + rand \times (X_{n,d}(t) - X_{r,d}(t)) \tag{11}$$

Selection and update stage: This is the stage where the most suitable individual is determined according to the fitness values of the two candidates, $X_{i-GWO}$ and $X_{i-DLH}$.

$$X_i(t+1) = \begin{cases} X_{i-GWO}(t+1), & iff\ (X_{i-GWO}) < f(X_{i-DLH}) \\ X_{i-DLH}(t+1) & otherwise \end{cases} \tag{12}$$

After all, these operations, if the fitness value of $X_i(t+1)$ obtained is better than $X_i(t)$, the value of $X_i(t+1)$ is updated as the new position using Eq. (12). Otherwise, $X_i(t)$ remains unchanged. After all of these processes have been applied for each candidate solution in the population, the number of iterations is increased by one and the process repeats until the termination conditions are met (*Nadimi-Shahraki, Taghian & Mirjalili, 2021*).

### New improved grey wolf optimizer (NI-GWO)

In this study, the NI-GWO algorithm, which is a new version of the I-GWO algorithm proposed by *Akyol, Yildirim & Alatas (2023)* was used. As seen in Eq. (3) in the I-GWO algorithm, the random number $r_1$ is used when calculating the $A$ coefficient. To enhance the algorithm performance, a value that linearly decreases from *alpha* to 0 given in Eq. (13) is used instead of a random number. In this study, 2 was chosen as the constant *alpha* value and *MaxIter* represents the maximum iteration number.

$$r_1 = alpha - t \times (alpha/MaxIter). \tag{13}$$

## Chaotic maps

Generating long-term random sequences with good uniformity is essential in sampling, numerical analysis, decision-making, and especially metaheuristic optimization. Successful generation of random numbers reduces computation and storage time to obtain the desired accuracy (*Cheng, Tran & Cao, 2016*). Recently, random number sequences have been replaced by chaotic number sequences and provide better performance in many complex problems. The use of chaotic number sequences has increased due to their spread spectrum properties, theoretical unpredictability, irregularity, stochastic properties, and related ergodic properties (*Akyol, Yildirim & Alatas, 2022*). In the literature, some studies increase performance by adding chaotic maps to optimization algorithms. The integrated version of the chaotic maps into the Optics Inspired Optimization algorithm was used in the deception detection problem (*Bingol & Alatas, 2023*). Tent chaotic maps are embedded to increase the initial population diversity of the Bald Eagle Search Optimization method (*Shen et al., 2022*). The Bernoulli chaotic map is integrated into the Sparrow Search Algorithm to improve the richness and diversity of the population (*Wang et al., 2022*).

### Circle map

Circle map, one of the one-dimensional chaotic maps, was defined by *Zheng (1994)*. This map, which belongs to the family of dynamical systems on the circle, is calculated as given in Eq. (14).

$$z_{m+1} = z_m + y_2 - \frac{y_1}{2\pi} \sin(2\pi z_m) \tag{14}$$

Here, $m$ is the generated chaotic number and $z_m$ is the $m$ th chaotic number in the range [0,1]. The $y_1$ and $y_2$ parameters also take the values of 0.5 and 0.2, respectively (*Lucay & Jamett, 2021*).

### Logistic map

The logistic map, introduced by May (2004), is one of the simplest one-dimensional chaotic maps and is described as shown in Eq. (15) (*May, 1976*; *Lucay & Jamett, 2021*).

$$z_{m+1} = \mu z_m (1 - z_m) \tag{15}$$

Here, the $\mu$ value is taken as 4 (*Lucay & Jamett, 2021*).

### Iterative map

Like the logistic map, Iterative Map was introduced by May. It is an Iterative Chaotic Map with infinite collapse and is expressed as shown in Eq. (16).

$$z_{m+1} = \sin\left(\frac{\gamma\pi}{z_m}\right) \tag{16}$$

where $\gamma \in (0,1)$ is a suitable parameter (*May, 1976*; *Gandomi et al., 2013*). The initial value of $z_1$ is taken as 0.7 for all three chaotic maps.

# PROPOSED METHODS

## Proposed chaotic NI-GWO algorithms

The convergence feature of NI-GWO is affected due to its stochastic nature, which uses a random number sequence for parameters during operation. As with other evolutionary computation-based algorithms, NI-GWO has no analytical results dependent on a specific number generator, which guarantees performance improvements. In this study, chaotic maps that emerge in nonlinear dynamics of biological populations with chaotic behavior are used for randomly generated numbers, and in this way, it is aimed to increase the effectiveness of NI-GWO. Based on this, four different chaotic NI-GWO algorithms are proposed using chaotic maps for the depression detection problem.

### CNI-GWO1

The initiation step of NI-GWO can affect the success of convergence. Thus, various chaotic systems are used rather than random number sequences in the start step of NI-GWO. Therefore, a chaotically initiated NI-GWO termed chaotic NI-GWO (CNI-GWO1) is proposed in this study. In this way, the global convergence of NI-GWO is tried to be improved and kept away from being trapped in local solutions. Equation (17) is used instead of Eq. (8), which is used to create the initial population in the NI-GWO algorithm.

$$X_{ij} = l_j + z_{m+1} \times (u_j - l_j), i \in [1,N], j \in [1,D] \tag{17}$$

The variable $z_{m+1}$ is calculated using Eqs. (14), (15) and (16). The pseudocode of the initial step is given in Fig. 3 and the flowchart of the CNI-GWO1 algorithm is shown in Fig. 4.

### CNI-GWO2

Second, in this proposed method, the randomly generated $C_1$ value in the range [0, 2] in Eq. (5) in NI-GWO is generated with chaotic maps. The resulting new expression is defined in Eq. (18).

$$D_\alpha = |(2 \times z_{m+1}) - X_\alpha - X(t)| \tag{18}$$

### CNI-GWO3

Third, in CNI-GWO3, the randomly generated $C_2$ value used for the hunting phase of the $\beta$ wolf in Eq. (5) is generated with chaotic maps. The resulting new expression is defined

```
Set initial values of z₁ = 0.7 and m = 1
   for i = 1 to PopulationNumber do
             for j = 1 to ProblemDimension do
                    Calculate z_{m+1} chaotic number based on the chaotic map used
                    X_{ij} = l_j + z_{m+1} × (u_j − l_j)
                    m = m + 1
             end for
   end for
```

**Figure 3** The pseudocode of the initial step.

in Eq. (19).

$$D_\beta = |(2 \times z_{m+1}) - X_\beta - X(t)| \tag{19}$$

### CNI-GWO4

Finally, in Eq. (5), chaotic maps are used instead of the $C_3$ value used to improve the hunting skills of the $\delta$ wolf. The updated version of the equation is given in Eq. (20).

$$D_\delta = |(2 \times z_{m+1}) - X_\delta - X(t)| \tag{20}$$

The flowchart of CNI-GWO2, CNI-GWO3, and CNI-GWO4 algorithms are shown in Fig. 5.

## Proposed chaotic intelligent optimization-based model for depression detection

In this study, depression detection is handled as a search problem. The chaotic NI-GWO algorithm is proposed to solve this problem. The KNN classifier and feature selection were used as components of the fitness function. An overview of the proposed chaos-integrated optimization model for the problem of depression detection is demonstrated in Fig. 6. The initial data set is separated as a train and test set for the models. Then, necessary text preprocessing and feature engineering are performed. NI-GWO and four proposed chaotic NI-GWO algorithms are adapted as new models for the interested problem. Accuracy from KNN and the number of features are simultaneously optimized for the classification model metrics. In the model created with the document matrix and optimization algorithms, the data allocated for testing was then classified and the performance of the model was measured. Furthermore, different types of popular supervised machine learning algorithms are also used for performance comparison of the proposed models. Accuracy, F-measure, MCC, sensitivity, and precision metrics are used to assess the success of proposed optimization-based models and machine learning algorithms.

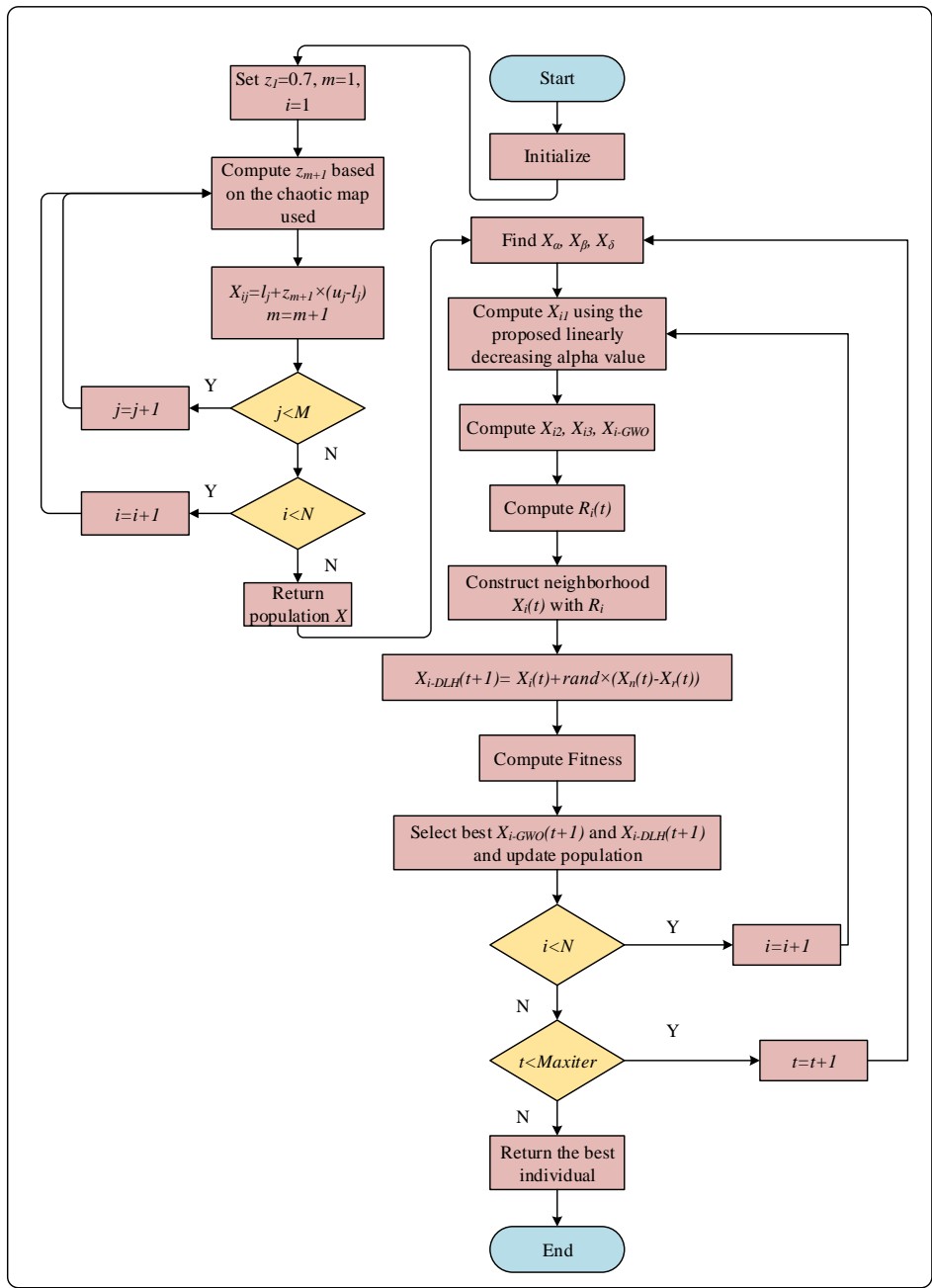

**Figure 4** The flowchart of CNI-GWO1 algorithm.

## Preprocessing of the depression dataset

In this study, the dataset created by labeling the data from Reddit (*Daru, 2023*) as "depressive" or "non-depressive" was used. This dataset consists total of 4,737 data, of which 3,937 are depressed and 900 are "non-depressed". This dataset is divided into

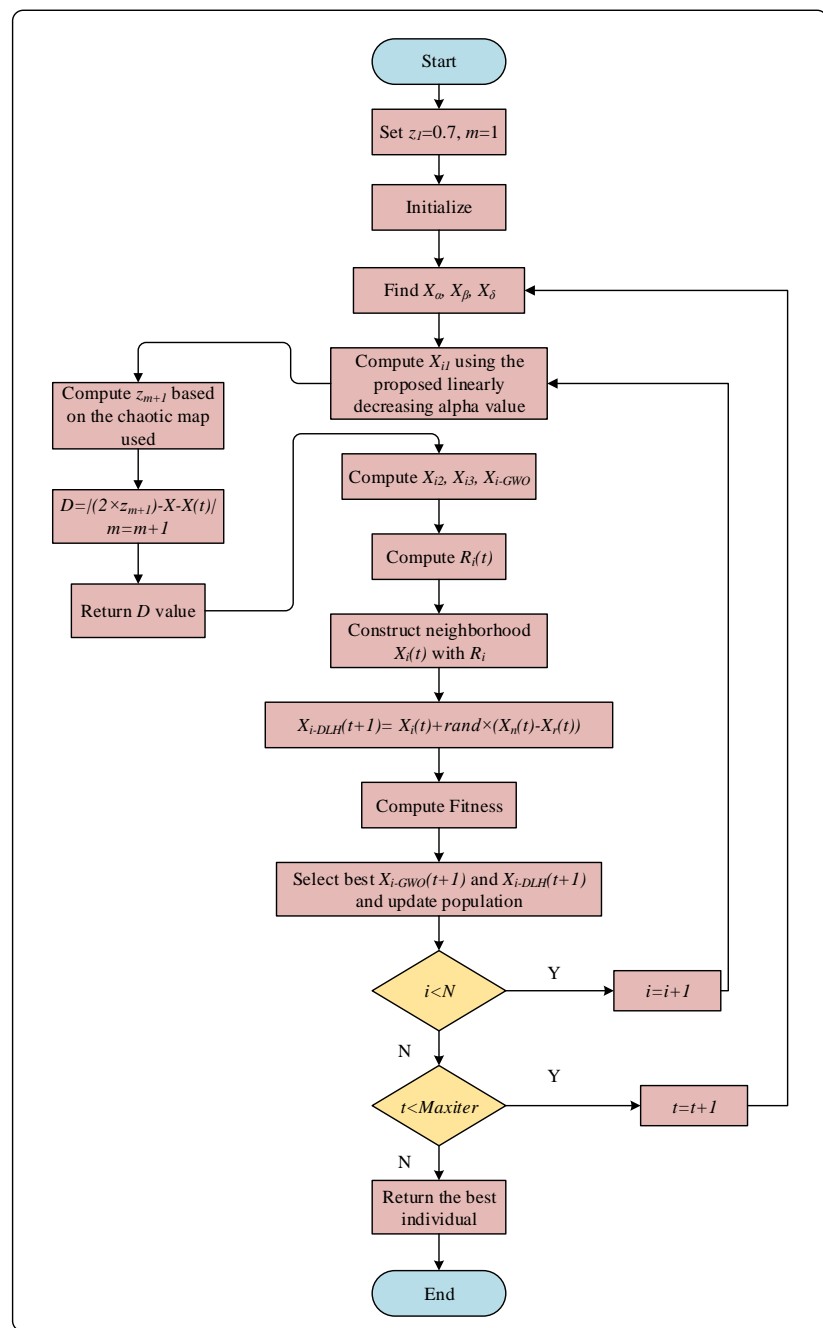

**Figure 5  The flowchart of CNI-GWO2, CNI-GWO3, and CNI-GWO4 algorithms.**

two separate datasets, 80% training and 20% testing. A section from this dataset is shown in Fig. 7.

The dataset used in this study was preprocessed and a document matrix was obtained. The number of words extracted from the dataset because of preprocessing was 1,117. In the document matrix, the rows represent the comments, while the columns correspond to

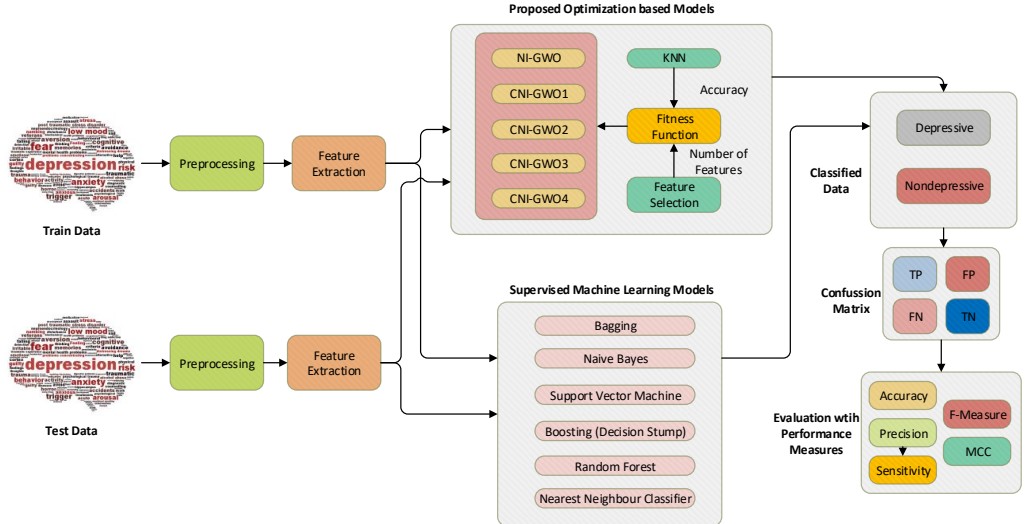

**Figure 6** Proposed chaos-integrated optimization model for depression detection.

| No. | 1: text String | 2: Class Nominal |
|---|---|---|
| 783 | Hey Reddit  What s your reason to be happy today ? | 0 |
| 784 | Forty-four years old and I am finally diagnosed as having anxiety. Well duh! It took you guys this long... | 0 |
| 785 | Found a potential life partner and I couldn t be happier! | 0 |
| 786 | Feeling lost loved ones at my cousin s wedding | 0 |
| 787 | Imagine a lowlife being able to overcome  traumas  depression  ADHD  OCD  confusion  drugs  alc... | 0 |
| 788 | My teacher was very encouraging to me  I m very happy | 0 |
| 789 | Today someone complimented my outfits!!! | 0 |
| 790 | Guys I m so sad can you cheer me up with a good joke :) | 0 |
| 791 | Told my favorite store clerk hes amazing and I think it made both our days | 0 |
| 792 | This letter to members of the LGBTQ+ community from a church in Austin  TX | 0 |
| 793 | I had a good birthday  something I ve not had in a long while | 0 |
| 794 | Be happy | 0 |
| 795 | I made myself smile today  and hopefully a few others too | 0 |
| 796 | One of my favorite youtubers responded to a comment I made on her video | 0 |
| 797 | I (19F) am having so much luck in my life!! I am so happy!!! | 0 |
| 798 | help | 1 |
| 799 | Hi | 1 |
| 800 | I feel empty. | 1 |
| 801 | How do I let go of what I grasp without intent? | 1 |
| 802 | Too much at a time | 1 |
| 803 | Pretending I m not depressed | 1 |
| 804 | How to tell people I m depressed? | 1 |
| 805 | I lied to my mom about applying to college | 1 |
| 806 | Dying now doesn t seem too bad now | 1 |
| 807 | I have no motivation and it s not good for the people around me | 1 |
| 808 | dying | 1 |
| 809 | Everytime I look at myself I can feel my heart drop. I really hate my face. | 1 |
| 810 | I m learning | 1 |
| 811 | Not depressed but I watched Up | 1 |

**Figure 7** A section from depression dataset.

the words in all comments. If the word is mentioned in the comment on the relevant line, it takes the value 1, if not, it takes the value 0. A section of the document matrix is shown in Fig. 8.

| 59: before | 60: beginning | 61: being | 62: believe | 63: best | 64: better | 65: between | 66: big | 67: birthday |
|---|---|---|---|---|---|---|---|---|
| Numeric | Numeric | Numeric | Numeric | Numeric | Numeric | Numeric | Numeric | Numeric |
| 0.0 | 0.0 | 0.0 | 0.0 | 0.0 | 0.0 | 0.0 | 0.0 | 0.0 |
| 1.0 | 0.0 | 0.0 | 0.0 | 0.0 | 0.0 | 0.0 | 0.0 | 0.0 |
| 0.0 | 0.0 | 0.0 | 0.0 | 0.0 | 0.0 | 0.0 | 0.0 | 0.0 |
| 0.0 | 0.0 | 0.0 | 0.0 | 0.0 | 0.0 | 0.0 | 0.0 | 0.0 |
| 0.0 | 0.0 | 0.0 | 1.0 | 1.0 | 1.0 | 0.0 | 0.0 | 0.0 |
| 0.0 | 0.0 | 1.0 | 0.0 | 0.0 | 0.0 | 0.0 | 0.0 | 0.0 |
| 0.0 | 0.0 | 1.0 | 0.0 | 0.0 | 0.0 | 0.0 | 0.0 | 0.0 |
| 0.0 | 0.0 | 0.0 | 0.0 | 0.0 | 0.0 | 0.0 | 0.0 | 0.0 |
| 0.0 | 0.0 | 0.0 | 0.0 | 0.0 | 0.0 | 0.0 | 0.0 | 0.0 |
| 0.0 | 0.0 | 0.0 | 0.0 | 1.0 | 1.0 | 0.0 | 0.0 | 0.0 |
| 0.0 | 0.0 | 0.0 | 0.0 | 0.0 | 0.0 | 0.0 | 0.0 | 0.0 |
| 0.0 | 0.0 | 0.0 | 0.0 | 0.0 | 0.0 | 0.0 | 0.0 | 0.0 |
| 0.0 | 0.0 | 0.0 | 0.0 | 0.0 | 0.0 | 0.0 | 0.0 | 0.0 |
| 0.0 | 0.0 | 0.0 | 0.0 | 0.0 | 0.0 | 0.0 | 0.0 | 0.0 |
| 1.0 | 0.0 | 0.0 | 0.0 | 1.0 | 0.0 | 0.0 | 0.0 | 0.0 |
| 0.0 | 0.0 | 0.0 | 0.0 | 0.0 | 0.0 | 0.0 | 0.0 | 0.0 |
| 0.0 | 0.0 | 0.0 | 0.0 | 0.0 | 0.0 | 0.0 | 1.0 | 0.0 |
| 0.0 | 0.0 | 0.0 | 0.0 | 0.0 | 0.0 | 0.0 | 1.0 | 0.0 |
| 0.0 | 0.0 | 0.0 | 0.0 | 1.0 | 0.0 | 0.0 | 0.0 | 0.0 |

**Figure 8** A section of the document matrix.

### Representation type (encoding)

The position vector of gray wolves for the focused depression detection problem is represented as follows:

$$X_{N,M} = \begin{bmatrix} X_{1,1} & X_{1,2} & \cdots & X_{1,M} \\ X_{2,1} & X_{2,2} & \cdots & X_{2,M} \\ \vdots & \vdots & \ddots & \vdots \\ X_{N,1} & X_{N,2} & \cdots & X_{N,M} \end{bmatrix}$$

In this encoding type, $N$ is the number of wolves and $M$ is used for representing the number of features extracted from the depression dataset. In other words, M represents the number of words extracted from the dataset, and N represents the number of population.

### Fitness function

While calculating the fitness of each individual in the population, the accuracy value obtained from the KNN classifier was used. In addition, a feature selection process is performed so that only the necessary features are included in the model. Thus, a multiobjective approach is used. The fitness function consisting of the combination of these two objectives is calculated as shown in Eq. (21).

$$fitness = w_1 \times AccuracyObtainedfromKNN + w_2 \times FeaturesRatio \tag{21}$$

Here, AccuracyObtainedfromKNN gives the accuracy rate obtained from the KNN classifier. FeaturesRatio gives the ratio of the number of features in the model to the total number of extracted features in the dataset. $w_1$ and $w_2$ represent the weights and their sum is equal to 1. In this study, the $w_1$ value was taken as 0.9 and the $w_2$ value as 0.1. Due to the direct usage of the selected features for classification, the proposed models seem to have

explainable capability. While creating the model, it is aimed to include only the necessary features in the model with feature selection. In the fit function, the number of features to be included in the model is kept low with FeaturesRatio. It is known how many features are included in the proposed model and which of these features are. In this way, the model has the characteristics of being explainable, interpretable, and comparable.

## Usability of the proposed method in live tweets

The proposed model in this article can be easily adapted to live or real-time tweets. After the live Tweets are preprocessed, they can be classified using the constructed model. However, real-time Tweet data are very large and in this manner, a Hadoop-based framework that allows the user to acquire and store tweets in a distributed environment and process them for detecting sarcastic content in real-time using the MapReduce programming model can be used. The mapper class can work as a partitioner and divides a large volume of tweets into small chunks and distributes them among the nodes in the Hadoop cluster. The reducer class can work as a combiner and can be responsible for collecting processed tweets from each node in the cluster and assembling them to produce the final output. Furthermore, due to the unknown class label of the Tweets, the performance of this system can be checked using other online tools.

## EXPERIMENTAL RESULTS

In the study, the results were obtained using MATLAB 2021b environment. Accuracy, precision, sensitivity, F-measure, and MCC criteria were used to evaluate the results obtained. First, the proposed CNI-GWO1, CNI-GWO2, CNI-GWO3, and CNI-GWO4 algorithms were evaluated according to the chaotic maps used in them. Because of running each algorithm 20 times, the best result, mean result, worst result, and the number of features obtained from the best result are given. In the algorithms, the population number is taken as 30 and the number of iterations as 100.

Table 2 shows the results of running the CNI-GWO1 algorithm 20 times, the best result, the worst result, the mean result, and the number of features obtained from the best result. These results show that the use of chaotic number sequences with spread spectrum properties, irregularity, stochastic properties, and related ergodic properties has improved the optimization-based model. Accordingly, the best results were obtained from the algorithms in which the initial population was created using logistic and iterative chaotic maps. According to the calculated mean results, it is seen that the best mean result is obtained from the algorithm using the circle map. The NI-GWO algorithm gave the worst result. Considering the number of features obtained in the best run, it is seen that the best result is obtained from the algorithm using the circle map with 526.

The results obtained by using circle, logistic, and iterative chaotic maps for the CNI-GWO2 algorithm are shown in Table 3. It can be seen that 100% accuracy is obtained using these three chaotic maps. It is clear that the best results are obtained from the integrated chaotic maps methods. Spread spectrum properties, stochastic properties, irregularity, and related ergodic properties of these maps also seem to be effective in obtaining these promising results. Based on the listed mean values, the best mean results are obtained from

**Table 2  Results obtained from CNI-GWO1.**

|  | Best | Mean | Worst | Number of Features from Best Solution |
|---|---|---|---|---|
| CNI-GWO1 with circle map | 0.9989 | 0.9987 | 0.9979 | 526 |
| CNI-GWO1 with logistic map | 1 | 0.9980 | 0.9968 | 553 |
| CNI-GWO1 with iterative map | 1 | 0.9983 | 0.9968 | 604 |
| NI-GWO | 0.9926 | 0.9909 | 0.9884 | 999 |

**Table 3  Results obtained from CNI-GWO2.**

|  | Best | Mean | Worst | Number of Features from Best Solution |
|---|---|---|---|---|
| CNI-GWO1 with circle map | 1 | 0.9989 | 0.9968 | 673 |
| CNI-GWO1 with logistic map | 1 | 0.9994 | 0.9989 | 699 |
| CNI-GWO1 with iterative map | 1 | 0.9998 | 0.9989 | 698 |
| NI-GWO | 0.9926 | 0.9909 | 0.9884 | 999 |

**Table 4  Results obtained from CNI-GWO3.**

|  | Best | Mean | Worst | Number of Features from Best Solution |
|---|---|---|---|---|
| CNI-GWO1 with circle map | 0.9989 | 0.9985 | 0.9979 | 680 |
| CNI-GWO1 with logistic map | 1 | 0.9983 | 0.9979 | 611 |
| CNI-GWO1 with iterative map | 1 | 0.9996 | 0.9989 | 665 |
| NI-GWO | 0.9926 | 0.9909 | 0.9884 | 999 |

the circle and iterative chaotic maps. Concerning the number of features obtained from these algorithms, it is seen that the least number of features is taken from the version using the circle map.

Table 4 provides the outcomes of the circle, logistic, and iterative chaotic maps integrated with the CNI-GWO3 algorithms. When this table is examined, 100% accuracy was obtained from the test data by integrating logistic and iterative chaotic maps into NI-GWO. Based on the mean values presented, the iterative chaotic map produces the best mean results. In terms of the number of features retrieved from them, the version employing the logistic map has the fewest.

The results of the chaotic maps integrated CNI-GWO4 algorithms are summarized in Table 5. Examining these data reveals that incorporating all three chaotic maps into the NI-GWO achieves 100% accuracy. The circle chaotic map yields the best mean results based on the supplied mean values. In terms of the number of features retrieved, it is evident that the circle map version has the fewest.

The proposed models are compared with popular supervised machine learning algorithms such as bagging, naive bayes, support vector machine, boosting (decision

**Table 5  Results obtained from CNI-GWO4.**

|  | Best | Mean | Worst | Number of Features from Best Solution |
|---|---|---|---|---|
| CNI-GWO1 with circle map | 1 | 0.9994 | 0.9979 | 594 |
| CNI-GWO1 with logistic map | 1 | 0.9983 | 0.9968 | 709 |
| CNI-GWO1 with iterative map | 1 | 0.9992 | 0.9979 | 641 |
| NI-GWO | 0.9926 | 0.9909 | 0.9884 | 999 |

stump), random forest, and nearest neighbour classifier. According to the result presented in *Aleem et al. (2022)*, these machine learning algorithms are the most widely used and successful methods in depression detection problems. In addition to machine learning algorithms, current metaheuristic algorithms Dandelion Optimizer (DO) (*Zhao et al., 2022*), Artificial Bee Colony Algorithm (ABC) (*Karaboga, 2010*), Harris Hawks Optimization (HHO) (*Heidari et al., 2019*), Sine Cosine Algorithm (SCA) (*Mirjalili, 2016*), Firefly Algorithm (FA) (*Yang, 2010*), and Bat Algorithm (BA) (*Yang, 2011*) comparison was also made. These metaheuristic algorithms used in the comparison were run 20 times, and the values of the best results obtained as a result of these 20 runs are presented in the figure. Confusion matrices obtained from the models based on the test depression data set are depicted in Fig. 9. In each confusion matrix, "0" represents "non-depressive" classes, and "1" represents "depressive" classes. Values in these confusion matrices show that; while worse results are yielded from supervised machine learning algorithms, better results are achieved from the proposed methods in this unbalanced data set. While the metaheuristic algorithms used in the comparison give better results than machine learning algorithms, it is seen that they give worse results than the proposed CNI-GWO algorithms.

Performance metric values computed according to the confusion matrices from all algorithms are presented in Table 6. CNI-GWO1 and CNI-GWO3 with logistic and iterative maps and CNI-GWO2 and CNI-GWO4 algorithms with all chaotic maps achieved the best accuracy, precision, sensitivity, F-measure, and MCC metrics with a value of 1. CNI-GWO1 and CNI-GWO3 with Circle Map seem to be the second-best methods in terms of accuracy, sensitivity, F-measure, and MCC metrics. Considering the results, it can be concluded that the proposed optimization-based models performed better than the different types of popular machine learning algorithms with respect to all evaluation metrics. The reason for the high performance of the proposed and adapted optimization algorithms is the distributed population-based search capability. Compared to other metaheuristic algorithms used in the comparison, it is seen that the proposed CNI-GWO algorithms give better results. Generally speaking, it seems that successful results are obtained in the use of metaheuristic algorithms in detecting depression.

Box plots of the results obtained in depression detection of DO, ABC, HHO, SCA, FA, and BA, which were used to compare the results with the CNI-GWO algorithms proposed in this study, are shown in Fig. 10. The graph shows the lower quartile, upper quartile, and median values obtained by running each algorithm 20 times. When the figure is examined, the lower quartile, upper quartile, and median values are close to each other

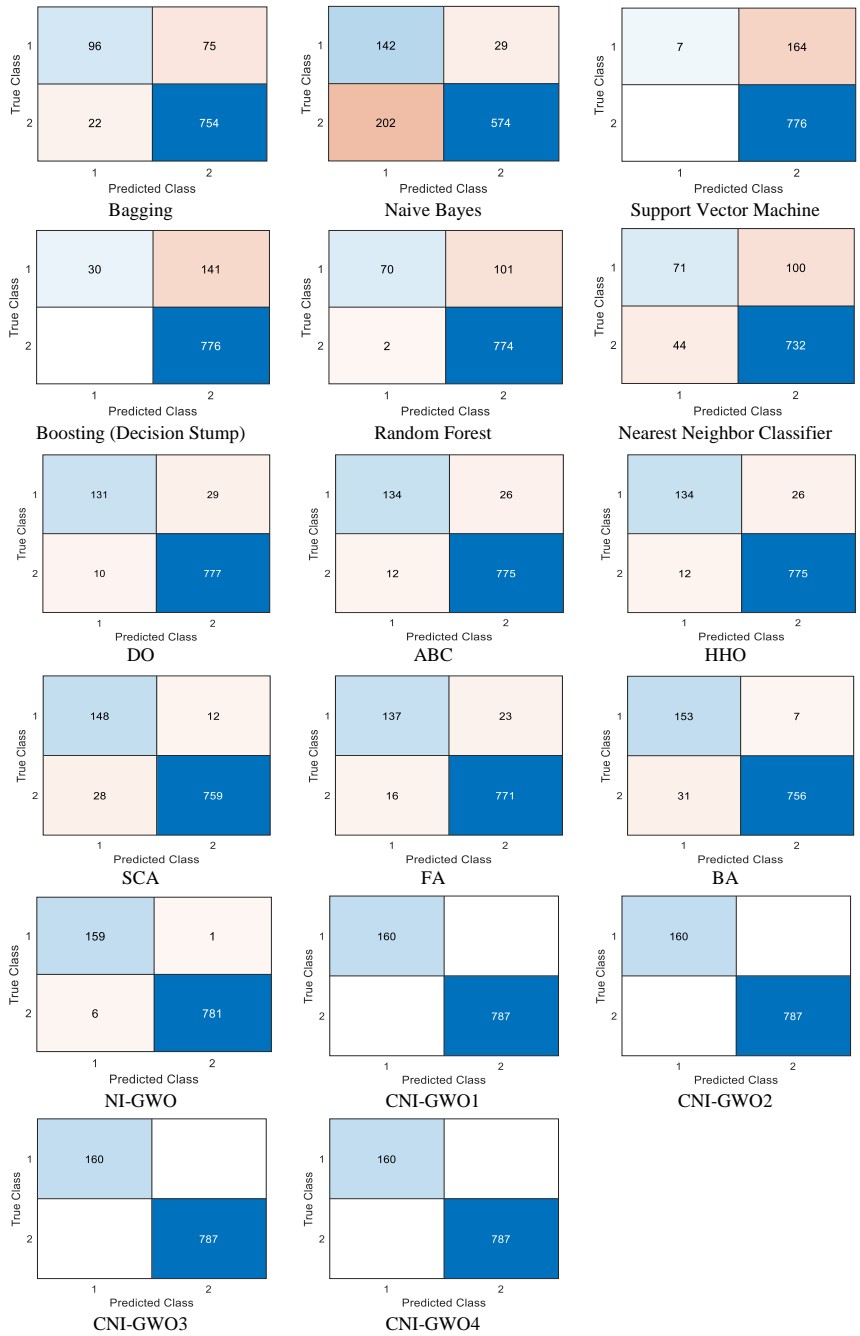

**Figure 9   Confusion matrices.**

in the proposed CNI-GWO algorithms. This shows that the algorithm performs well in each run. In addition, when compared to other algorithms, it is seen that the algorithms proposed in this study give better results than other algorithms.

Finally, the experimental results were evaluated with the non-parametric Friedman test to examine whether there was a statistically significant difference between the results

**Table 6  Performance metrics of the algorithms.**

|  | Accuracy | Precision | Sensitivity | F-Measure | MCC |
|---|---|---|---|---|---|
| Bagging | 0.8976 | 0.8920 | 0.8980 | 0.8900 | 0.6210 |
| Naive Bayes | 0.7561 | 0.8550 | 0.7560 | 0.7820 | 0.4560 |
| Support Vector Machine | 0.8268 | 0.8570 | 0.8270 | 0.7550 | 0.1840 |
| Boosting (Decision Stump) | 0.8511 | 0.8740 | 0.8510 | 0.8050 | 0.3850 |
| Random Forest | 0.8912 | 0.9000 | 0.8910 | 0.8720 | 0.5900 |
| Nearest Neighbour Classifier | 0.8479 | 0.8320 | 0.8480 | 0.8360 | 0.4220 |
| DO | 0.9588 | 0.9291 | 0.8187 | 0.8704 | 0.8485 |
| ABC | 0.9599 | 0.9178 | 0.8375 | 0.8758 | 0.8532 |
| HHO | 0.9599 | 0.9122 | 0.8438 | 0.8766 | 0.8536 |
| SCA | 0.9578 | 0.8409 | 0.9250 | 0.8810 | 0.8568 |
| FA | 0.9588 | 0.8954 | 0.8562 | 0.8754 | 0.8511 |
| BA | 0.9599 | 0.8315 | 0.9563 | 0.8895 | 0.8683 |
| NI-GWO | 0.9926 | 0.9940 | 0.9938 | 0.9742 | 0.9740 |
| CNI-GWO1 with circle map | 0.9989 | 0.9938 | 1 | 0.9962 | 0.9962 |
| CNI-GWO1 with logistic map | 1 | 1 | 1 | 1 | 1 |
| CNI-GWO1 with iterative map | 1 | 1 | 1 | 1 | 1 |
| CNI-GWO2 with circle map | 1 | 1 | 1 | 1 | 1 |
| CNI-GWO2 with logistic map | 1 | 1 | 1 | 1 | 1 |
| CNI-GWO2 with iterative map | 1 | 1 | 1 | 1 | 1 |
| CNI-GWO3 with circle map | 0.9989 | 0.9938 | 1 | 0.9962 | 0.9962 |
| CNI-GWO3 with logistic map | 1 | 1 | 1 | 1 | 1 |
| CNI-GWO3 with iterative map | 1 | 1 | 1 | 1 | 1 |
| CNI-GWO4 with circle map | 1 | 1 | 1 | 1 | 1 |
| CNI-GWO4 with logistic map | 1 | 1 | 1 | 1 | 1 |
| CNI-GWO4 with iterative map | 1 | 1 | 1 | 1 | 1 |

obtained from the metaheuristic algorithms used in this study. The Friedman test is used to identify differences in the behavior of multiple algorithms. Here, the sample size shows how many results are obtained from each algorithm, and the median value shows the average value of these results. There are two different hypotheses in this test: null hypothesis (*H0*) and alternative hypothesis (*H1*). In the experiments, *H0* means "There is no significant difference between the fitness function values of the compared algorithms and the fitness function values of the proposed CNI-GWO". *H1* means "There is a significant difference between the fitness function values of the compared algorithms and the fitness function values of the proposed CNI-GWO". The results of the analysis performed with the Friedman test are shown in Table 7. The alpha value was determined as 0.05 and degrees of freedom (*Df*) (number of samples compared -1) was determined as 7. According to the alpha value and degrees of freedom, it can be seen from the chi-square distribution table that the $x_F^2$ value is 14.067. According to this table, it can be seen that the $x^2$ value is greater than the $x_F^2$ value ($x^2 > x_F^2$). Accordingly, *H0* is rejected and *H1* is accepted. In other words, there is a statistically significant difference between the fitness function values of the compared

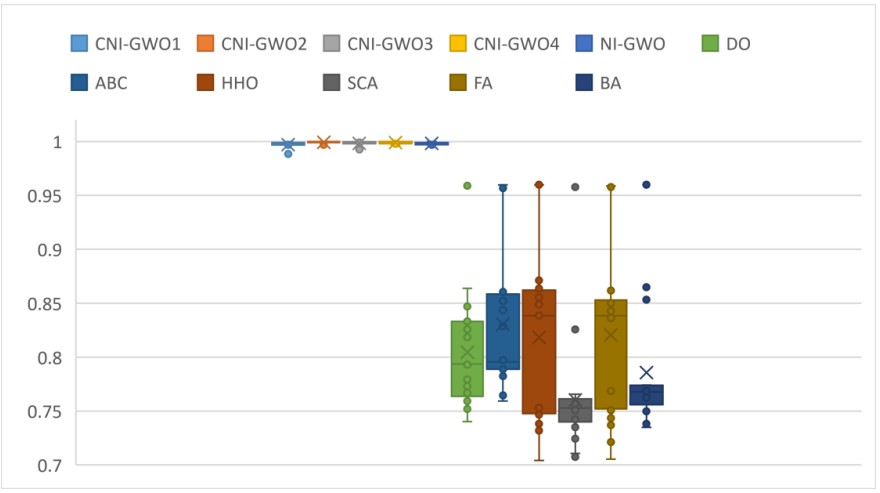

**Figure 10  Box plots for experimental results of metaheuristic algorithms.**

**Table 7  Analysis of the results with the Friedman test.**

|  | CNI-GWO | NI-GWO | DO | ABC | HHO | SCA | FA | BA |
|---|---|---|---|---|---|---|---|---|
| **Sample size** | 20 | 20 | 20 | 20 | 20 | 20 | 20 | 20 |
| **Median** | 0.9989 | 0.9984 | 0.7936 | 0.7957 | 0.8384 | 0.7529 | 0.8384 | 0.7677 |
| **Sum of ranks** | 154 | 146 | 73.5 | 88 | 74 | 41 | 76 | 67.5 |
| **Mean of the ranks** | 7.7 | 7.3 | 3.675 | 4.4 | 3.7 | 2.05 | 3.8 | 3.375 |
| **$\chi^2$-score** | 90.832836 | | | | | | | |
| **Degrees of freedom (statistics)** | 7 | | | | | | | |
| **$p$-value** | 6.66E−16 | | | | | | | |

**Notes.**
The result is very siginificant at $p < 0.05$.

algorithms and the fitness function values of the proposed CNI-GWO according to the Friedman test.

In general, looking at the results, it can be seen that the proposed CNI-GWO algorithms perform better than both machine learning algorithms and other metaheuristic algorithms in detecting depression. Looking at the Confusion matrices in Fig. 9, it can be seen that the proposed algorithms have a very high number of correct labeling of the data, even though it is an unbalanced dataset. Similarly, looking at Table 6, it is seen that the proposed CNI-GWO algorithms achieved a 100% success rate in accuracy, precision, sensitivity, F-measure, and MCC metrics. While other metaheuristic algorithms perform well around 95% in the Accuracy metric, this success rate is around 80% in other metrics. Machine learning algorithms gave worse results. Finally, a non-parametric Friedman test was performed to see whether the proposed algorithm creates a semantic difference from other

metaheuristic algorithms used in comparison. Looking at the test results, it can be seen that the proposed algorithm has a statistically significant difference. Based on these, it seems that metaheuristic algorithms achieve successful results in detecting depression. It has been observed that this success rate increases by integrating chaotic maps.

## CONCLUSIONS

Depression is a psychological impact of modern-day living on people's minds. Identification, prevention, and treatment of depression as well as the determination of relapse risk factors are critical because depression is a serious individual and social health problem due to the risk of suicide and loss of workforce, as well as high chronicity, recurrence rates and prevalence. In this study, new chaos-integrated optimization algorithms are proposed for automatic depression detection. The topic of detecting depression in online social media and networks is approached as an optimization problem for the first time. The problem search area involves social media and network data, and novel chaos-integrated improved optimization methods are applied as solution search strategies and models for automatic depression detection. Besides, in this study, a new version of the I-GWO algorithm with chaotic maps integrated is proposed. To increase the success rate, an appropriate encoding type and flexible multi-objective fitness function that can simultaneously optimize accuracy and the number of features is suggested.

Based on the results, it can be concluded that the proposed optimization-based models outperformed popular machine learning algorithms across accuracy, precision, sensitivity, F-measure, and MCC evaluation metrics. In the proposed models, 100% success was achieved in accuracy, precision, sensitivity, F-measure, and MCC values. In other current optimization algorithms used for comparison, it is seen that more than 90% success is achieved for the accuracy value. In machine learning algorithms, it is seen that it has a success rate of 75% to 89%. The distributed multipoint-based search capacity plays a part in the outstanding performance of the proposed optimization methods. The expression multipoint-based search refers to a search process starting from multiple points. This method helps to obtain optimal metric values in a social media dataset where the data is unbalanced, without increasing the computational complexity. Furthermore, although the used social media data are imbalanced, optimum metric values are reached without any additional process that increases the computational complexity. Spread spectrum properties, stochastic properties, irregularity, and ergodic properties of the chaotic maps also seem to be effective in obtaining these promising results. In addition, according to confusion matrices, it can be seen that supervised machine learning algorithms performed worse distributions, while proposed chaotic optimization-based models performed better. Because they consist of reduced features that directly represent the related words themselves; constructed depression detection models have explainable, comprehensible, and interpretable capabilities that differ from traditional black-box methods. One of the limitations of the study is the limited number of data in the study. Another limitation is the absence of specialist doctors in the study. In future studies, it is aimed to work with more multicenter data and more experts in the field.

For future studies, adaptive and hybrid versions of the intelligent optimization algorithms can be proposed for better performance in automatic depression detection. The proposed explainable multiobjective metaheuristic-based models can also be used for other social media and network analysis problems. Distributed programming paradigms can be integrated to use this model in big social streaming online data. New different objectives can be added to the current ones, and Pareto-based multiobjective methods can be proposed for simultaneously optimizing conflicting and contradicting criteria.

### Funding
The authors received no funding for this work.

### Competing Interests
The authors declare there are no competing interests.

### Author Contributions
- Sinem Akyol conceived and designed the experiments, performed the experiments, analyzed the data, performed the computation work, prepared figures and/or tables, authored or reviewed drafts of the article, and approved the final draft.

### Data Availability
The code is available in the Supplemental File.

The data is available at Kaggle: https://www.kaggle.com/datasets/deapdaru/depression-detection-from-reddit. DOI: https://doi.org/10.1007/978-981-99-1414-2_44. Collaborators: Deap Daru (Owner). Authors: Deap Daru, Hitansh Surani, Harit Koladia, Kunal Parmar.

### Supplemental Information
Supplemental information for this article can be found online at http://dx.doi.org/10.7717/peerj-cs.1661#supplemental-information.

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
