# Peer review of "New chaos-integrated improved grey wolf optimization based models for automatic detection of depression in online social media and networks"

_PeerJ Computer Science, doi:10.7717/peerj-cs.1661_

## Round 0.1 · original submission · Major Revisions

Dear authors

Your paper has been reviewed by the experts in their field and you will see that they have a couple of suggestions to improve your article,

Therefore, please revise carefully and resubmit.

**Language Note:** The review process has identified that the English language must be improved. PeerJ can provide language editing services - please contact us at copyediting@peerj.com for pricing (be sure to provide your manuscript number and title). Alternatively, you should make your own arrangements to improve the language quality and provide details in your response letter. – PeerJ Staff

Reviewer 1 ·

Basic reporting

I request authors to pay more attention to English proofreading strategy. There are more grammar errors and language errors in the model.

Experimental design

Algorithms and text analysis procedures must be discussed in detail. Also, there are more abbreviations used in the heading. Rewrite the errors.

Validity of the findings

In line 34, “algorithmsare” has a spacing error. The sentence formation error needs to be resolved in the article.
It might be helpful, what techniques used in the research can be discussed with novelty in the abstract. It helps readers to understand the research novelty easily.
There is no quantitative discussion made in the abstract. The last few lines regarding the resulting values need to be added in the abstract.
Add the motivation behind this research in the introduction section.
line 147, sentence formation errors are noted. Rewrite the sentence.
Heading GWO should not be used as heading 3.1. reconsider the heading.
Algorithms and text analysis procedures must be discussed in detail. Also, there are more abbreviations used in the heading. Rewrite the errors.
All the figures must be cited In the article. Also, add descriptions to the figures in detail.
Add result discussion in section 4, at the end of the results.
rewrite the conclusion with more details on the proposed novelty and achievements compared to existing models.

·

Basic reporting

In the study, the authors propose new chaos-integrated multi-objective optimization algorithms in order to overcome the limitations of studies on depression detection problem by increasing the success rate and automatically selecting related features and integrating explainability. The results of their proposed algorithms are compared with different types of popular supervised machine learning algorithms that are widely and successfully used for the depression detection problem. The paper seems original and interesting results are presented. Some of the concerns, questions, and advices about the paper are listed below:

1. How the proposed model performs “explainable, comprehensible, and interpretable” capabilities is not clearly described.
2. Adding a few sentences focusing more on the contributions of the paper and major outcomes of the study in the Abstract section will be better.
3. Beside contributions, it is better to include the main limitations of the research in the conclusion.
4. For better readability, the authors may expand the abbreviations at every first occurrence.
5. The “Conclusions” are a key component of the paper. It should complement the “Abstract” and normally used by experts to value the paper’s engineering content. In general, it should sum up the most important outcomes of the paper. It should simply provide critical facts and figures achieved in this paper for supporting the claims.
6. What are the real-life use cases of the proposed model? The authors can add a theoretical discussion on the real-life usage of the proposed model.
7. “Adolescent use of digital and social mediums has changed…” should be changed. Correct usage of “mediums” and “media” is important.
8. A reference should be given after “Previous research has found that automatic text-based analysis of depressive symptoms can be used to detect depressing phrases or disrespectful sentences in blog posts or conversations as well as sentiment recovery from suicide notes.” sentence.
9. On page 6 in line 146: Blank character should be correctly used.
10. The motivation on integration chaos into optimization is not clearly explained in “Introduction” section.
11. Equation (1) is the same as Equation (3). They must be corrected.
12. Equation (2) is the same as Equation (4). They must be corrected.
13. Numbering and citation of all equations should be checked.
14. Encircling phase of GWO seems to be written twice. Please see line 270 and 277 on page 9.
15. In line 281 of page 10, although “a”, “r1”, and “r2” are explained, they do not seem in the equations.
16. In line 322 and 323 of page 11; although “… coefficient a and the coefficients A and C are computed by Equation 3-5.” is written, these calculation of these coefficients are not seen in these equations.
17. Equation (12) should be correctly written.
18. “After all, of these processes have been applied for each candidate solution in the population” should be corrected as “After all of these processes have been applied for each candidate solution in the population,”.
19. Correct equation number should be used “As seen in Equation 3 in the I-GWO algorithm, the random number r1 is used when calculating the A coefficient” or the parameters should be correctly written.
20. “(May, 1976, Gandomi et al., 2013].” should be corrected.
21. Number of Equation (17) should be correctly written.
22. “The variable z_(m+1) is” in line 422 should be correctly written.
23. “…the randomly generated C_1 value in the range…” in line 430, “…the randomly generated C_2 value…” in line 436, and “chaotic maps are used instead of the C_3 value used to improve” in line 443 should be corrected.
24. In line 486 on page 15, N and M in sentence “…N is the number of wolves and M is…” should be written in italic as in the equations.
25. In line 496 and 497, variables (AccuracyObtainedfromKNN and FeaturesRatio) should be written in italic.
26. In line 498 and 499, “w_1 and w_2 represent the weights and their sum is equal to 1. In this study, the w_1 value was taken as 0.9 and the w_2 value as 0.1.” should be corrected. “w_1 and w_2” do not seem in the equations.
27. In line 548, the author write “integrating the logistic and iterative chaotic maps into the NI-GWO yields 100% accuracy.”. It is not clear whether this accuracy value is obtained from train data or test data?
28. Name of the chaotic maps should be written correctly. Please use lower case or upper case but not both of them. See line 550, 556, 558, 577, etc.
29. “According to the result presented in [7],…” should be corrected. There is not any reference for “[7]”. Referencing style should be corrected.
30. What is “multipoint – based search” written in the “Conclusions” section?

Experimental design

The experimental design is reasonable and sufficient. It analyzes the performance of the proposed algorithm and compares it with the existing ones.

Validity of the findings

The paper evaluates the proposed algorithm based on sufficient experiments. Experimental results show the validity of the findings.

Reviewer 3 ·

Basic reporting

Dear Authors,
although proposed manuscript has merits, there are some issues that need to be addressed.

Introduction should be clearly presented to highlight main ideas and motivation behind the proposed research. Please include and clearly state research question and contributions of proposed study in Introduction. Also, please clearly explain what is "beyond state-of-the-art" in the proposed study.

Literature review should be improved to include more metaheuristics-based approaches, since the GWO belongs to the group of nature-inspired metaheuristics. In this context, a novel and prospective research field - hybrid methods between metaheuristics and machine learning. Consider using the following papers to enhance the literature review (you may just say that the novel research field successfully combines machine learning and swarm intelligence approaches and proved to be able to obtain outstanding results in different areas):

https://www.mdpi.com/1328174
https://link.springer.com/article/10.1007/s00521-018-3937-8

Experimental design

In general, experimental section is solid.
However, more state of the art (SOTA) baseline approaches should be included in comparative analysis. You have to include metaheuristics-based methods in comparative analysis, e.g. FA, SCA, BA, HHO, CS, ABC, FFO, etc.

Visualization of results should be improved - consider using box and whiskers diagrams, swarm plots, etc. It would be interesting to use swarm plots that depict population diversity in the final iteration of the best run.

Validity of the findings

To prove the significance of obtained results, statistical tests must be conducted. There are many statistical tests appropriate for validating metaheuristics results, please choose some tests from the following reference: https://www.sciencedirect.com/science/article/pii/S2210650211000034

Additional comments

Conclusion should be extended to include more details regarding the future work and limitations of proposed study. Limitations should be distinguished between theoretical and practical.

Some references are missing parts, such as pp., publisher, year, etc.

There are some English language and technical errors, please revise, e.g. tables must be aligned on the center of the page, all symbols in equations must be defined, etc.

---

## Round 0.2 · accepted · Accept

Thank you very much for your revision and fine contribution to our esteemed journal.

Reviewer 1 ·

Basic reporting

Authors have included all the corrections pointed by the reviewers.

Experimental design

In the revised version, authors improved experimental results and presentation of main contributions.

Validity of the findings

In the revised version, authors improved experimental results and presentation of main contributions.

Additional comments

In the revised version, authors improved experimental results and presentation of main contributions. I have no other comments in the next round.

·

Basic reporting

All my questions are well answered and the paper quality is improved. I have no further concern and recommend as accepted.

Experimental design

The experimental design is clear and reasonable.

Validity of the findings

The findings is efficient and novel.

Reviewer 3 ·

Basic reporting

Authors have revised their manuscript properly.

Experimental design

Authors have revised their manuscript properly.

Validity of the findings

Authors have revised their manuscript properly.

Additional comments

Authors have revised their manuscript properly.